# UNI-IR: AMBIGUITY-REDUCED INVERSE RENDERING THROUGH A UNIFIED FRAMEWORK FOR GLOSSY OBJECTS

## ABSTRACT

Inverse rendering aims to decompose the an image into geometry, materials, and lighting. Recently, Neural Radiance Fields (NeRF) based inverse rendering has significantly advanced, bridging the gap between NeRF-based models and conventional rendering engines. Existing methods typically adopt a two-stage optimization approach, beginning with volume rendering for geometry reconstruction, followed by physically based rendering (PBR) for materials and lighting estimation. However, the inherent ambiguity between materials and lighting during PBR and the suboptimal nature of geometry reconstruction by volume rendering only compromise the outcomes. To address these challenges, we introduce Uni-IR, a unified framework that imposes mutual constraints to alleviate ambiguity by integrating volume rendering and physically based rendering. Specifically, we employ a physically-based volume rendering (PBVR) approach that incorporates PBR concepts into volume rendering, directly facilitating connections with materials and lighting, in addition to geometry. Both rendering methods are utilized during optimization, imposing mutual constraints and optimizing geometry, materials, and lighting synergistically. Employing a meticulously crafted unified representation for both lighting and materials, Uni-IR achieves high-quality geometry reconstruction, materials and lighting estimation across various object types.

## 1 INTRODUCTION

Multi-view 3D reconstruction is a pivotal task in computer vision and computer graphics, serving as a cornerstone for various applications such as game modeling Gregory (2018); Lewis & Jacobson (2002), computer animation Parent (2012); Lasseter (1987), and virtual reality Schuemie et al. (2001). Despite the remarkable progress achieved by Neural Radiance Fields (NeRF) Mildenhall et al. (2021) and subsequent approaches like SDF-based neural implicit surface learning Wang et al. (2021); Yariv et al. (2021); Oechsle et al. (2021), multi-view 3D reconstruction still presents challenges in bridging the gap between NeRF-based models and conventional rendering engines. Volume rendering, the core mechanism of NeRF, generates radiance without explicitly considering the interactions of materials and lighting. In contrast, conventional rendering engines derive shading through the interactions between surface materials Nicodemus (1965) and lighting.

To bridge the gap, inverse rendering, the task of disentangling radiance into geometry, materials, and lighting, has garnered significant attention. This approach allows the reconstructed 3D model to be directly integrated into rendering engines, thereby playing a critical role in downstream applications such as game production Lewis & Jacobson (2002) and virtual reality Schuemie et al. (2001).

Recent studies Zhang et al. (2022); Sun et al. (2023); Zhang et al. (2021a;b); Liu et al. (2023); Yang et al. (2023) have explored the inverse rendering task within a neural implicit surface learning framework. These approaches typically adopt a two-stage training strategy. In the initial stage, volume rendering is utilized for geometry reconstruction. Subsequently, in the second stage, physically based rendering is employed to refine materials and lighting under a fixed geometry as shown in Figure 1 (a). However, the performance in the second stage heavily relies on the quality of geometry reconstruction in the first stage. While these methods demonstrate the effectiveness on the objects with diffuse materials, they often lead to suboptimal results in reconstructing reflective surfaces,

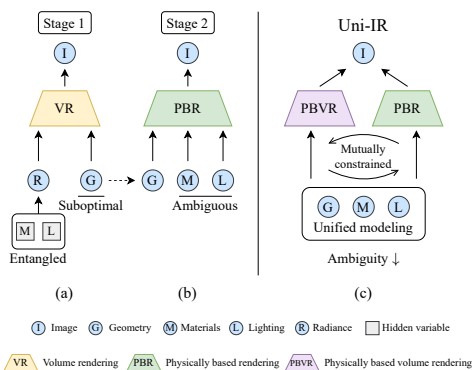 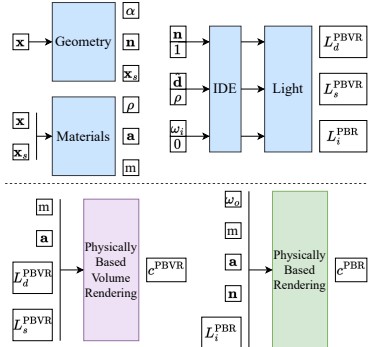

Figure 1: Two-stage and unified optimization.  Figure 2: An overview of Uni-IR framework.

consequently leading to compromised materials and lighting predictions in the second stage. More recently, several studies have focused on object reconstruction with reflections Ge et al. (2023); Verbin et al. (2022); Liu et al. (2023); Liang et al. (2023a), with some of them Liu et al. (2023); Liang et al. (2023a) further leveraging well-reconstructed geometry for inverse rendering. Despite the promising results achieved, the ill-posed nature of inverse rendering makes optimization still challenging. Lack of constraints in physically based rendering leads to suboptimal performance, with inherent ambiguity between lighting and materials in Figure 1 (b).

To tackle the challenges, we propose to integrate volume rendering and physically based rendering to simultaneously optimize geometry, materials, and lighting, aiming at imposing mutual constraints to alleviate the ambiguity. While this idea appears straightforward, unifying the two rendering pipelines is non-trivial. Merely integrating the two into a unified framework will not introduce mutual constraints, since volume rendering lacks physical plausibility and severs the connection with physically based rendering. Conversely, the optimization becomes intractable due to the ambiguity among geometry, materials, and lighting. To address this issue, we depart from the traditional volume rendering, which employs an MLP to predict radiance with entangled materials and lighting as shown in Figure 1 (a). Instead, we integrate the principles of PBR into volume rendering inspired by Liu et al. (2023). In this approach, radiance is also formulated as the interaction between the materials and lighting. We refer to this method as physically based volume rendering (PBVR).

By integrating PBVR and PBR within a unified framework, we introduce a method named Uni-IR as illustrated in Figure 1 (c). At the heart of this approach lies a meticulously crafted unified representation for materials and lighting representations across both rendering methods, achieving mutual constraints and optimizing geometry, materials, and lighting synergistically. In the lighting representation, we consolidate the lighting representation for both PBVR and PBR through integrated directional encoding Verbin et al. (2022). Two distinct light MLPs are employed to model direct and indirect lighting, respectively, addressing global illumination and inter-reflection, which are crucial for differentiating indirect light from albedo and environment map. Additionally, we parameterize the visibility term through two distinct MLPs. One is probabilistic for PBVR, while the other is deterministic for PBR, indicating whether direct or indirect lights should be used. For materials prediction, we employ a shared material MLP for both PBVR and PBR.

To summarize, our contributions are listed as follows.

- To the best of our knowledge, we introduce the first unified optimization framework for inverse rendering task, seamlessly integrating PBVR and PBR. Unifying these two rendering methods synergistically enhances the performance of inverse rendering.

- We meticulously design a unified lighting and materials representation for PBVR and PBR, effectively imposing mutual constraints and mitigating ambiguity.

- We present a comprehensive evaluation protocol, encompassing assessments of geometry reconstruction, as well as material and lighting estimation accuracy. Extensive experiments conducted on multiple datasets demonstrate the effectiveness of the proposed framework.

## 2 RELATED WORK

### 2.1 MULTI-VIEW 3D RECONSTRUCTION

Traditional multi-view 3D reconstruction typically employs Multi-View Stereo (MVS) techniques Curless & Levoy (1996); Fua & Leclerc (1995); Furukawa & Ponce (2009); Schönberger et al. (2016); Schonberger & Frahm (2016); Xu & Tao (2019), with the objective of reconstructing scene geometry from multi-view images. These techniques utilize multi-view consistency to establish correspondences and estimate depth values across different views, yielding a point cloud reconstruction. However, MVS methods encounter challenges in reconstructing reliable geometry in specific scenarios, such as surfaces with specular reflections and regions with low texture. In such cases, multi-view consistency is compromised, potentially resulting in severe artifacts and missing parts in the reconstructed output.

With the recent advancements in deep learning, learning-based approaches utilizing implicit surface representations have emerged, where neural networks are employed to map continuous points to either an occupancy field Mescheder et al. (2019); Peng et al. (2020) or a Signed Distance Function (SDF) Park et al. (2019). Unlike traditional methods, these approaches are immune to appearance changes as they rely on 3D ground truth supervision. However, these methods typically necessitate additional supervision corresponding to the occupancy value or SDF for each point. Unfortunately, such supervision may not always be readily available when utilizing solely multi-view 2D images, thereby restricting their scalability.

The advent of volumetric approaches in NeRF Mildenhall et al. (2021) has sparked significant interest in 3D reconstruction utilizing neural implicit surface representations Oechsle et al. (2021); Yariv et al. (2021); Wang et al. (2021). Subsequent research endeavors have continued to enhance the reconstruction performance across various aspects. Despite demonstrating promising performance in 3D reconstruction, these methods still struggle to accurately recover the geometry of specular surfaces. Therefore, in this study, we concentrate on the reconstruction of reflective objects, along with the estimation of material and lighting properties.

### 2.2 MODELING FOR OBJECT WITH REFLECTION

Recently, several studies Boss et al. (2021); Srinivasan et al. (2021); Zhang et al. (2021a;b); Verbin et al. (2022); Ge et al. (2023); Liu et al. (2023); Wang et al. (2024; 2023); Fan et al. (2023) have focused on modeling objects with reflection. Some studies Verbin et al. (2022); Boss et al. (2021); Zhang et al. (2021b); Bi et al. (2020); Rodriguez et al. (2020); Kopanas et al. (2022); Liang et al. (2023b) tackle rendering tasks, where they model view-dependent reflective appearances by decomposing a scene into shape, reflectance, and illumination for novel view synthesis and relighting. Other studies Ge et al. (2023); Liu et al. (2023); Wang et al. (2024); Fan et al. (2023); Liang et al. (2023a) focus on reconstructing reflective geometry by modeling specular light more reasonably or mitigating the effects of specular surfaces. For example, Ref-NeuS Ge et al. (2023) and NeP Wang et al. (2024) reduce the impact of highly uncertain reflective regions while enhancing the significance of less altered areas. Additionally, ENVIDR Liang et al. (2023a) and NeRO Liu et al. (2023) adopt a more physically plausible approach to model specular color, leading to notable improvements in reconstruction performance. Typically, physically based rendering is employed to further estimate materials and lighting given well-reconstructed geometry. Despite achieving promising results, the entanglement between illumination and materials compromises outcomes due to the ill-posed nature of inverse rendering, inevitably leading to suboptimal results in many cases. In this study, we propose a unified framework that imposes mutual constraints and alleviates the entanglement issue.

### 2.3 INVERSE RENDERING WITH NEURAL IMPLICIT LEARNING

Inverse rendering Barron & Malik (2014); Nimier-David et al. (2019) aims to decompose image appearance into intrinsic properties such as geometry, materials, and lighting. This task has posed a challenge in computer vision and graphics due to its ill-posed nature. Recovering reliable intrinsic properties is particularly difficult because of the limited constraints added during optimization. To address this challenge, most existing methods Zhang et al. (2022); Liu et al. (2023); Yu et al. (1999); Zhang et al. (2021b); Yao et al. (2022); Wang et al. (2024); Yang et al. (2023) employ a

geometry-first optimization framework. Initially, they utilize volume rendering to reconstruct geometry. Subsequently, in the second stage, physically based rendering is used for materials and lighting estimation. However, ambiguity between materials and lighting hinders the second-stage optimization. These methods do not establish a connection between volume rendering and physically based rendering, as they are performed separately in two distinct stages. Our approach integrates PBVR, which incorporates the concept of PBR into traditional volume rendering to establish a connection with PBR, and PBR within a unified framework. Featuring a carefully designed unified representation for both lighting and material representation, our method effectively imposes mutual constraints and mitigates ambiguity.

## 3 APPROACH

With $N$ calibrated multi-view images denoted as $\mathcal{X} = \{\mathbf{I}_i\}_{i=1}^N$, our objective is to address the inverse rendering that simultaneously reconstructing of the object's geometry and estimating the materials and lighting. We commence by providing a succinct overview of volume rendering and physically based rendering in Section 3.1. Next, we introduce physically based volume rendering (PBVR), and how we integrate PBR and PBVR into a unified framework in Section 3.2. Subsequently, we delve into the design of unified lighting and materials representations in Section 3.3. Lastly, Section 3.4 presents full optimization. An overview of our framework is illustrated in Figure 2.

### 3.1 PRELIMINARIES

**Volume Rendering.** Volume rendering Kajiya & Von Herzen (1984) used in NeRF Mildenhall et al. (2021) aims at multi-view 3D reconstruction and novel view synthesis. The core idea is to represent the continuous attributes (i.e., density and radiance) of a 3D scene with neural networks. $\alpha$ compositing Max (1995) aggregates these attributes along a ray $\mathbf{r}$ to approximate the pixel RGB values by:

$$\hat{C}(\mathbf{r}) = \sum_{i=1}^{P} T_i \alpha_i \boldsymbol{c}_i, \tag{1}$$

where $T_i = \exp\left(-\sum_{j=1}^{i-1} \alpha_j \delta_j\right)$ and $\alpha_i = 1 - \exp\left(-\sigma_i \delta_i\right)$ denote the transmittance and alpha value of sampled point, respectively. $\delta_i$ is the distance between neighboring sampled points. $P$ is the number of sampled points along a ray. $\sigma_i$ and $\boldsymbol{c}_i$ are predicted attributes by the neural networks. The training object $\mathcal{L}$ is the mean square error between the ground-truth pixel color $\boldsymbol{C}(\mathbf{r})$ and the rendered color $\hat{C}(\mathbf{r})$ formulated as

$$\mathcal{L}_{\text{render}} = \sum_{\mathbf{r} \in \mathcal{R}} \|\boldsymbol{C}(\mathbf{r}) - \hat{C}(\mathbf{r})\|_2^2, \tag{2}$$

where $\mathcal{R}$ is the set of all rays shooting from the camera center to image pixels. Subsequent approaches use Signed Distance Function (SDF) instead opacity $\sigma$ to define the geometry. Following NeuS Wang et al. (2021), the formulation of $\alpha_i$ is calculated from the signed distance $g(\boldsymbol{x})$ rather than density $\sigma_i$ as

$$\alpha_i = \max\left(\frac{\Phi_s\left(g(\boldsymbol{x}_i)\right) - \Phi_s\left(g(\boldsymbol{x}_{i+1})\right)}{\Phi_s\left(g(\boldsymbol{x}_i)\right)}, 0\right), \tag{3}$$

where $g$ is the geometry network, which maps a position $\boldsymbol{x}$ to its signed distance $g(\boldsymbol{x})$. $\Phi_s(x) = (1 + e^{-sx})^{-1}$ and $1/s$ is a trainable parameter which indicates the standard deviation of $\Phi_s(x)$.

**Physically Based Rendering.** Physically based rendering aims to produce photo-realistic 2D images given geometry, materials and lighting. At its core, the rendering equation Kajiya (1986) models the interaction between materials and lighting in a physically plausible manner. It inherently represents an integral equation that describes the equilibrium of light in a scene. The formula is expressed as

$$\boldsymbol{c}(\boldsymbol{x}, \boldsymbol{\omega_o}) = \int_\Omega f(\boldsymbol{x}, \boldsymbol{\omega_o}, \boldsymbol{\omega_i}) L_i(\boldsymbol{x}, \boldsymbol{\omega_i})(\boldsymbol{\omega_i} \cdot \mathbf{n}) d\boldsymbol{\omega_i}, \tag{4}$$

where $\boldsymbol{\omega_o}$ is the viewing direction of the outgoing light, $L_i$ is the incident light of direction $\boldsymbol{\omega_i}$ sampled from the upper hemisphere $\Omega$ of the surface point $\boldsymbol{x}$, and $\mathbf{n}$ is the surface normal. $f$ is the

BRDF properties. The function $f$ consists of a diffused and a specular component

$$f(\boldsymbol{x}, \boldsymbol{\omega_o}, \boldsymbol{\omega_i}) = (1-m)\frac{\boldsymbol{a}}{\pi} + \frac{DFG}{4(\boldsymbol{\omega_i} \times \mathbf{n})(\boldsymbol{\omega_o} \times \mathbf{n})}, \tag{5}$$

where $m \in [0,1]$ is the metallic of the surface point. $\boldsymbol{a} \in [0,1]^3$ is the albedo color of the point. $D$ is the normal distribution function, $F$ is the Fresnel term and $G$ is the geometry term, which are all determined by the metallic m, the roughness r and the albedo $\boldsymbol{a}$. We detail the expression of $D$, $F$ and $G$ in the Appendix. With Eq. equation 4 and Eq. equation 5, the outgoing radiance is given by

$$\boldsymbol{c}(\boldsymbol{x}, \boldsymbol{\omega_o}) = \boldsymbol{c}_{\mathrm{d}}(\boldsymbol{x}, \boldsymbol{\omega}_o) + \boldsymbol{c}_{\mathrm{s}}(\boldsymbol{x}, \boldsymbol{\omega}_o), \tag{6}$$

$$\boldsymbol{c}_{\mathrm{d}}(\boldsymbol{x}, \boldsymbol{\omega_o}) = (1-m)\boldsymbol{a} \int_{\Omega} L_i(\boldsymbol{x}, \boldsymbol{\omega_i})\frac{(\boldsymbol{\omega_i} \cdot \mathbf{n})}{\pi}d\boldsymbol{\omega_i}, \tag{7}$$

$$\boldsymbol{c}_{\mathrm{s}}(\boldsymbol{x}, \boldsymbol{\omega_o}) = \int_{\Omega} \frac{DFG}{4(\boldsymbol{\omega_i} \times \mathbf{n})(\boldsymbol{\omega_o} \times \mathbf{n})}L_i(\boldsymbol{x}, \boldsymbol{\omega_i})(\boldsymbol{\omega_i} \cdot \mathbf{n})d\boldsymbol{\omega_i}. \tag{8}$$

## 3.2 Unifying Volume Rendering and Physically Based Rendering

It is not trivial to integrate volume rendering (VR) and physically based rendering (PBR) into a unified framework for simultaneous reconstructing the object's geometry and estimating its materials and lighting. A naive solution is to evaluate the rendering equation on surface points. However, materials and lighting remain only related to PBR without integrating with volume rendering, thus failing to introduce mutual constraints, since traditional volume rendering adopts an MLP for direct radiance prediction, entangling the materials and lighting. Consequently, the optimization process becomes intractable due to the ambiguity among geometry, materials, and lighting.

To guarantee that volume rendering also incorporates materials and lighting, we integrate it with the principles of PBR, which allows the radiance to be computed by modeling the interaction between materials and lighting. Inspired by NeRO Liu et al. (2023), we represent the radiance of each sampled point along a ray using a simplified rendering equation, which approximates the lighting with light MLP instead of integral in Eq. equation 7 and equation 8, termed PBVR. The diffuse and specular components are

$$\boldsymbol{c}_{\mathrm{d}}^{\mathrm{PBVR}}(\boldsymbol{x}, \boldsymbol{\omega_o}) = (1-m)\boldsymbol{a}L_{\mathrm{d}}^{\mathrm{PBVR}}, \quad L_{\mathrm{d}}^{\mathrm{PBVR}} \approx \int_{\Omega} L_i(\boldsymbol{x}, \boldsymbol{\omega_i})D(\boldsymbol{n}, 1)d\boldsymbol{\omega_i} \tag{9}$$

$$\boldsymbol{c}_{\mathrm{s}}^{\mathrm{PBVR}}(\boldsymbol{x}, \boldsymbol{\omega_o}) = m\boldsymbol{a}L_{\mathrm{s}}^{\mathrm{PBVR}}, \quad L_{\mathrm{s}}^{\mathrm{PBVR}} \approx \int_{\Omega} L_i(\boldsymbol{x}, \boldsymbol{\omega_i})D(\hat{\boldsymbol{d}}, \rho)d\boldsymbol{\omega_i} \tag{10}$$

where $L_{\mathrm{d}}^{\mathrm{PBVR}}$, $L_{\mathrm{s}}^{\mathrm{PBVR}}$ are the approximated diffuse and specular light, respectively. $D(\hat{\boldsymbol{d}}, \rho)$ is the normal distribution function (i.e., specular lobe), $\hat{\boldsymbol{d}}$ is the reflective direction. $D(\boldsymbol{n}, 1) \approx \frac{(\boldsymbol{\omega_i} \cdot \mathbf{n})}{\pi}$ is the diffuse lobe. We elaborate the simplified process from Eq. equation 8 to Eq. equation 10 in the Appendix.

For physically based rendering, we evaluate the rendering equation on the surface points $\boldsymbol{x}_s = \mathbf{o} + \mathbf{d}\sum_{i=1}^{P} T_i\alpha_i t_i$, where $\mathbf{o}$ is the camera origin, $\mathbf{d}$ is the camera direction and $t_i$ is the depth of $i$-th sampled point. We adopt Monte Carlo sampling to approximate the diffuse color and specular color. The diffused color is estimated by sampling $N_d$ rays with a cosine-weighted probability

$$\boldsymbol{c}_{\mathrm{d}}^{\mathrm{PBR}}(\boldsymbol{x}_s, \boldsymbol{\omega_o}) = (1-m)\boldsymbol{a}\sum_{i}^{N_d} L_i^{\mathrm{PBR}}, \tag{11}$$

where $i$ indicate the $i$-th sampled direction. For specular color, we adopt the GGX distribution as normal distribution $D$. We sample $N_s$ rays follows DDX distribution Cook & Torrance (1982) to estimate specular color

$$\boldsymbol{c}_{\mathrm{s}}^{\mathrm{PBR}}(\boldsymbol{x}_s, \boldsymbol{\omega_o}) = \frac{1}{N_s}\sum_{i}^{N_s} \frac{FG(\boldsymbol{\omega_o} \cdot \mathbf{h})}{(\mathbf{n} \cdot \mathbf{h})(\mathbf{n} \cdot \boldsymbol{\omega_o})}L_i^{\mathrm{PBR}}, \tag{12}$$

where $\mathbf{h}$ is the half-way vector between $\boldsymbol{\omega_i}$ and $\boldsymbol{\omega_o}$, $L_i^{\mathrm{PBR}}$ is predicted light of $i$-th sampled direction.

Both rendering methods require materials and lighting for shading. By integrating PBVR and PBR with a carefully crafted materials and lighting representation, we enhance the mutual constraints for inverse rendering, thereby reducing the probability of converging to suboptimal results.

### 3.3 Unifying Lighting and Materials Representation

All the radiance terms in Eq. equation 9, Eq. equation 10, Eq. equation 11, and Eq. equation 12 depend on lighting and materials. Appropriately representing these elements is essential for effectively imposing mutual constrains and mitigating ambiguity among geometry, materials, and lighting.

**Lighting Representation.** Given the crucial role that global illumination and inter-reflection play in distinguishing indirect light from albedo and environment maps, we utilize two distinct MLPs to separately encode direct and indirect lighting. The direct light MLP $l_{\text{direct}}(SH(\boldsymbol{\omega}_i))$ takes only direction as input, ensuring a globally consistent direct environment map. $SH(\cdot)$ is the directional encoding using spherical harmonics as basis functions. This model is applicable when the path from point $\boldsymbol{x}$ to direction $\boldsymbol{\omega}_i$ is unobstructed. In contrast, the indirect light MLP $l_{\text{indirect}}(SH(\boldsymbol{\omega}_i), \boldsymbol{x})$ requires both position and direction as input to accommodate the spatial variability of indirect lighting across the scene. This model is used when the path from $\boldsymbol{x}$ to $\boldsymbol{\omega}_i$ encounters obstructions.

To establish a unified lighting representation, we utilize integrated directional encoding (IDE) Verbin et al. (2022), which shows the integral of light in Eq. equation 9 and Eq. equation 10 has a closed-form solution by representing the $L_i(\boldsymbol{x}, \boldsymbol{\omega}_i)$ with spherical harmonics, based on direction and roughness denoted as $\text{IDE}(\boldsymbol{\omega}, \text{k})$. Although the roughness term in $\text{IDE}(\boldsymbol{\omega}, \rho)$ is defined by the von Mises-Fisher (vMF) distribution, which differs from the roughness term in the GGX distribution used in PBR, both serve similar functions by defining positively correlated concentration. We optimize the roughness as the parameter in the GGX distribution and use it for lighting approximation in PBVR.

For PBVR, the integrals of diffuse and specualr light can be approximated by

$$L_{\text{d}}^{\text{PBVR}} = l_{\text{direct}}\left(\text{IDE}(\boldsymbol{n}, 1)\right), \quad L_{\text{s}}^{\text{PBVR}} = l_{\text{direct}}(\text{IDE}(\hat{\boldsymbol{d}}, \rho)). \tag{13}$$

In PBR, the diffuse and specular light are both computed by

$$L_i^{\text{PBR}} = l_{\text{direct}}\left(\text{IDE}(\boldsymbol{\omega}_i, 0)\right), \tag{14}$$

where $\rho$ is set to 0 since sampled directions are deterministic instead of a distribution. For indirect light, the position $\boldsymbol{x}$ is additionally inputted. We consolidate the lighting representation for both PBVR and PBR, encompassing both specular and diffuse components. This consolidation effectively imposes constraints on lighting optimization and alleviates entanglement issues.

**Visibility Representation.** Given the inclusion of indirect light in our lighting representation, it is critical to estimate a visibility term to correctly apply direct or indirect light. In PBR, light is determined through Monte Carlo sampling, where each sampled direction is deterministic, resulting in binary visibility values of either 0 or 1, denoted as $v_i^{\text{PBR}} \in \{0, 1\}$. An MLP maps the surface point $\boldsymbol{x}_s$ and sampled direction $\boldsymbol{\omega}_i$ to visibility, defined as $v_i^{\text{PBR}} = \mathcal{V}^{\text{PBR}}(\boldsymbol{x}_s, \boldsymbol{\omega}_i)$. In PBVR, visibility is probabilistic, denoted as $v^{\text{PBVR}} \in [0, 1]$, since lighting representation in Eq. equation 13 approximates the specular light using a single direction and roughness. When roughness is large, the light integral is influenced not only by the reflective direction $\hat{\boldsymbol{d}}$. Thus, another MLP maps the sampled point $\boldsymbol{x}$ and $\text{IDE}(\hat{\boldsymbol{d}}, \rho))$ to visibility, denoted as $v^{\text{PBVR}} = \mathcal{V}^{\text{PBVR}}(\boldsymbol{x}, \text{IDE}(\hat{\boldsymbol{d}}, \rho))$. To account for the deterministic and probabilistic property, we use visibility by ray-marching in geometry network and visible proportion by Monte Carlo sampled directions as supervision, respectively. The visibility loss is given by

$$\mathcal{L}_{vis} = \|v_i^{\text{PBR}} - v_i^{\text{march}}\|_1 + \|v^{\text{PBVR}} - \frac{1}{N_s}\sum_{i=1}^{N_s} v_i^{\text{PBR}}\|_1. \tag{15}$$

Given the visibility, the light $L_{\text{s}}^{\text{PBVR}}$ in Eq. equation 13 can be expressed as

$$L_{\text{s}}^{\text{PBVR}} = v^{\text{PBVR}} l_{\text{direct}}(\text{IDE}(\hat{\boldsymbol{d}}, \rho)) + (1 - v^{\text{PBVR}}) l_{\text{indirect}}(\text{IDE}(\hat{\boldsymbol{d}}, \rho), \boldsymbol{x}). \tag{16}$$

Since diffuse light primarily contains low-frequency information, we do not explicitly model the indirect diffused light for PBVR. The light $L_i^{\text{PBR}}$ in Eq. equation 14 is modified as

$$L_i^{\text{PBR}} = v_i^{\text{PBR}} l_{\text{direct}}(\text{IDE}(\boldsymbol{\omega}_i, 0)) + (1 - v_i^{\text{PBR}}) l_{\text{indirect}}(\text{IDE}(\boldsymbol{\omega}_i, 0), \boldsymbol{x}_s). \tag{17}$$

**Materials Representation.** Material representation, including metallic $m$, roughness $\rho$, and albedo $\boldsymbol{a}$, is conducted using a material MLP $\mathcal{M}_{\text{material}}$ based on position $\boldsymbol{x}$, denoted as $\{m, \rho, \boldsymbol{a}\} = \mathcal{M}_{\text{material}}(\boldsymbol{x})$, and these predictions are shared across PBR-based rendering and PBR. The distinction lies in the fact that material prediction operates on ray-based points and surface-based points, respectively. This difference introduces two types of constraints for material optimization.

### 3.4 OPTIMIZING

During the training process, our total loss function is

$$\mathcal{L} = \mathcal{L}_{\text{render}}^{\text{PBVR}} + \lambda_{\text{PBR}} \mathcal{L}_{\text{render}}^{\text{PBR}} + \lambda_{\text{eik}} \mathcal{L}_{\text{eik}} + \lambda_{\text{vis}} \mathcal{L}_{\text{vis}} + \lambda_{\text{mat\_reg}} \mathcal{L}_{\text{mat\_reg}}, \tag{18}$$

where $\mathcal{L}_{\text{render}}$ is the Charbonier loss Barron et al. (2022) calculated between the rendered color and the ground-truth color. In PBVR, the rendered color is derived from Eq. equation 1, where each $c_i$ combines $c_{\text{d}}^{\text{PBVR}}$ and $c_{\text{s}}^{\text{PBVR}}$ as outlined in Eqs. 9 and 10, and $\alpha_i$ determined by Eq. equation 3. In the context of PBR, the rendered color is formulated as $C = c_{\text{d}}^{\text{PBR}} + c_{\text{s}}^{\text{PBR}}$, based on Eq. equation 11 and Eq. equation 12. $\mathcal{L}_{\text{eik}}$ is an eikonal term Gropp et al. (2020) to regularize the gradients of geometry network formualated as

$$\mathcal{L}_{\text{eik}} = \frac{1}{P} \sum_{i=1}^{P} \left( |\nabla f(\boldsymbol{x})| - 1 \right)^2. \tag{19}$$

$\mathcal{L}_{\text{mat\_reg}}$ is a smoothness regularization to ensure the material more smooth in the space

$$\mathcal{L}_{\text{mat\_reg}} = \|\mathcal{M}(\boldsymbol{x}_s) - \mathcal{M}(\boldsymbol{x}_s + \epsilon)\|_2, \tag{20}$$

where $\epsilon = 5e - 3$.

## 4 EXPERIMENTS

### 4.1 DATASETS AND EVALUATION PROTOCOL

To evaluate the effectiveness of our method, we conducted experiments on objects from several datasets. These include synthetic data from ShinyBlender Verbin et al. (2022) and CompoBlender, where objects are composed from ShinyBlender or Blender Mildenhall et al. (2021), featuring more complex scenes and inter-reflections (see Appendix for more details), as well as real captured data from Stanford-ORB Kuang et al. (2024).

We present a comprehensive evaluation protocol, encompassing assessments of geometry reconstruction accuracy as well as materials and lighting estimation accuracy.

**Geometry Reconstruction.** The evaluation metric used is the Chamfer Distance, provided by the DTU evaluation metrics Aanæs et al. (2016). This metric comprises two components: *accuracy* and *completeness*. Consistent with Ref-NeuS Ge et al. (2023), only accuracy is reported on ShinyBlender and CompoBlender. We also reported the results of Stanford-OBR in the same scale.

**Materials Estimation.** Given access to ground truth of albedo, roughness, and metallic maps for the synthetic datasets, Mean Squared Error (MSE) was reported for metallic and roughness, and PSNR was used for diffuse albedo. For the real dataset Stanford-ORB, where ground truth for roughness and metallic maps is unavailable, qualitative relighting results including PSNR, SSIM and LPIPS were provided as an alternative. Besides, pseudo albedo was used to evaluate predicted albedo.

**Lighting Estimation.** For lighting evaluation, akin to DeepLight LeGendre et al. (2019) and StyleLight Wang et al. (2022), we employ three spheres with different materials for assessment: mirror silver, matte silver, and diffuse grey, depicted in Figure 3. The three spheres are rendered with ground-truth lighting and the estimated environment map using Blender Hess (2013). Evaluation metrics include RMSE, scale-invariant RMSE (si-RMSE) and Angular Error.

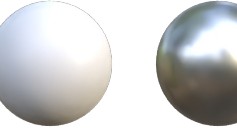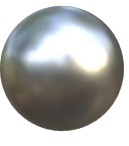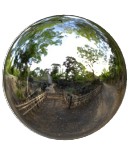

Gray Diffuse     Matte Silver     Mirror Silver

Figure 3: Three spheres with different materials: mirror silver, matte silver, and diffuse grey.

### 4.2 IMPLEMENTATION DETAILS

Our model was developed based on NeRO Liu et al. (2023). The architecture of the geometry network, lighting network, and material network mirrors that of NeRO. For more details, please see our Appendix. Our model underwent training for 200,000 iterations, requiring 12 hours on a single

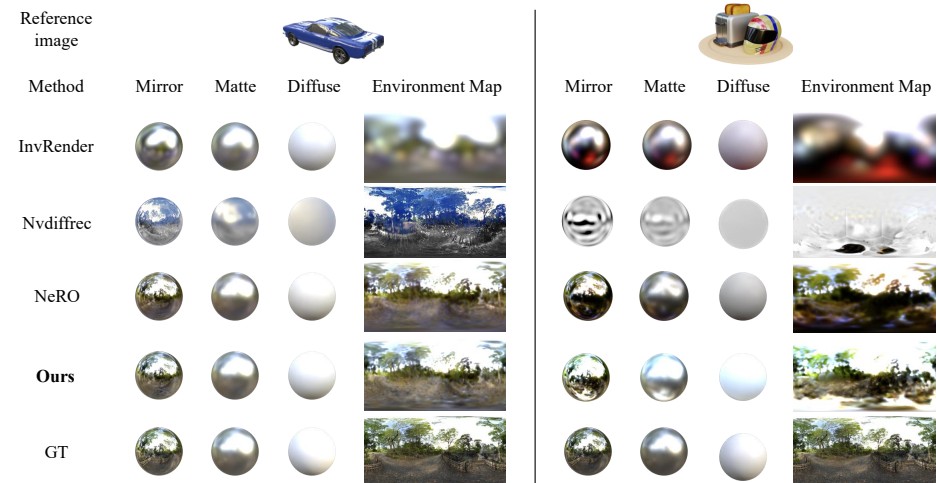

Figure 4: The environment map and rendered spheres on "car" and "toahel". We ran InvRender, Nvidiffrec, NeRO official implementations. Our method obviously produces better result.

NVIDIA RTX 3090 Ti GPU. Upon convergence, a mesh was extracted from the signed distance functions within a predefined bounding box using the Marching Cubes Lorensen & Cline (1987) at a resolution of 512. An environment map with a resolution of 512 × 1024 was generated by uniformly sampling across azimuth and elevation in spherical space, followed by querying the light using the direct light MLP. Note that although our approach builds upon NeRO Liu et al. (2023), we believe it can be adapted to any volumetric neural implicit framework. For example, techniques such as Instant-NGP and CUDA-based Monte Carlo sampling can be readily leveraged for acceleration.

## 4.3 COMPARISON WITH STATE-OF-THE-ART METHODS

We compared the results of our method with several other methods, including NeRO Liu et al. (2023), InvRender Zhang et al. (2022) and NvdiffRec Munkberg et al. (2022) and a 3D Gaussian Splatting based inverse rendering method GaussianShader (GShader) Jiang et al. (2024) on both synthetic dataset, and compared with cutting-edge method NeRO on real captured dataset. The quantitative results are shown in Tables 1 and 2. Since InvRender assume dielectric materials, the

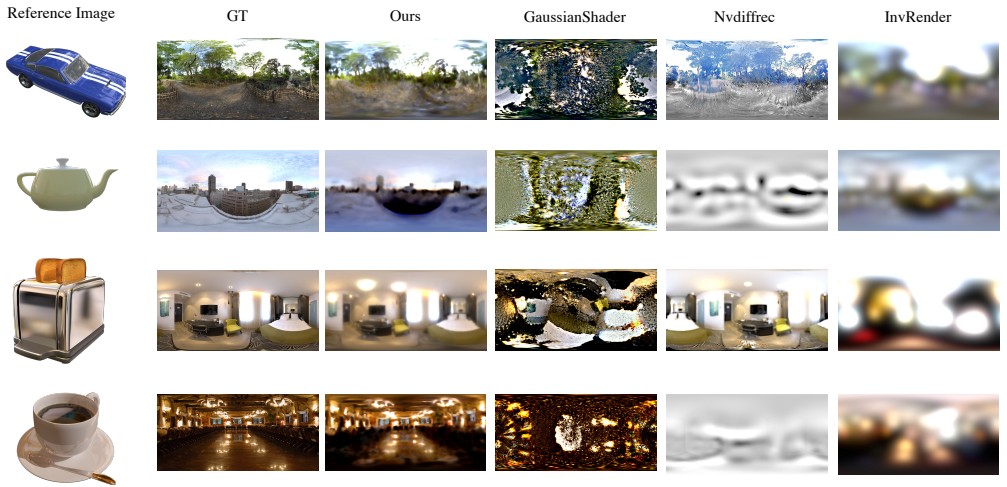

Figure 5: Visualization of estimated environment map. Our method can recover fine-grained environment map given only multi-view 2D images.

Table 1: Comparison with state-of-the-art methods on ShinyBlender and CompoBlender Dataset. **Bold** results have the best score. Our method outperforms these methods by a large margin.

| Method | ShinyBlender | | | CompoBlender | | |
|---|---|---|---|---|---|---|
| | Geometry | Materials | Lighting | Geometry | Materials | Lighting |
| GShader | 1.37 | 0.041 / 0.069 / 20.10 | 13.71 / 2.24 / 2.09 | 1.69 | 0.125 / 0.101 / 16.01 | 12.35 / 2.21 / 1.52 |
| Nvdiffrec | 2.59 | 0.045 / 0.074 / 19.90 | 14.11 / 2.30 / 2.05 | 2.95 | 0.138 / 0.110 / 15.93 | 10.70 / 2.12 / 1.41 |
| InvRender | 1.39 | 0.035 / - / - | 11.38 / 2.09 / 2.04 | 1.35 | 0.069 / - / - | 15.09 / 2.40 / 1.90 |
| NeRO | 0.67 | 0.023 / 0.030 / 22.26 | 8.86 / 1.65 / 1.97 | 2.05 | 0.063 / 0.055 / 17.68 | 9.40 / 1.73 / 1.04 |
| Ours | **0.58** | **0.015 / 0.025 / 23.21** | **7.91 / 1.58 / 1.45** | **0.82** | **0.039 / 0.026 / 18.84** | **8.17 / 1.54 / 0.88** |

Table 2: Comparison with cutting-edge method NeRO on Stanford-ORB Dataset. **Bold** results have the best score. Our method performs better on real captured dataset.

| Method | Geometry | Relighting | | | Material |
|---|---|---|---|---|---|
| | CD ↓ | PNSR ↑ | SSIM ↑ | LPIPS ↓ | PSNR ↑ |
| NeRO | 1.35 | 25.45 | 0.898 | 0.054 | 23.25 |
| Ours | **0.97** | **26.13** | **0.902** | **0.051** | **24.84** |

metallic and diffuse albedo are not available. We reported the mean result for each evaluation metric. For lighting, we further averaged the results on three spheres. Please refer to our Appendix for more details. Our method significantly outperforms all other compared methods on all evaluation metrics. We shown the qualitative comparison of lighting estimation in Figure 4 and the extracted 2D environment map by querying the optimized direct light MLP in Figure 5. We also visualized the qualitative comparison of geometry reconstruction in Figure 6. Note that though the 3D Gaussian Splatting-based method Jiang et al. (2024) excels in optimization speed, its performance is significantly inferior. More visualizations are in the Appendix, where we also discuss how incorrect materials can lead to wrong geometry reconstruction.

## 4.4 ABLATION STUDY

We conducted an ablation study on the "coffee" object from ShinyBlender and the "gnome" object from Stanford-ORB to evaluate the effectiveness of unifying PBR and PBVR, as well as the unified

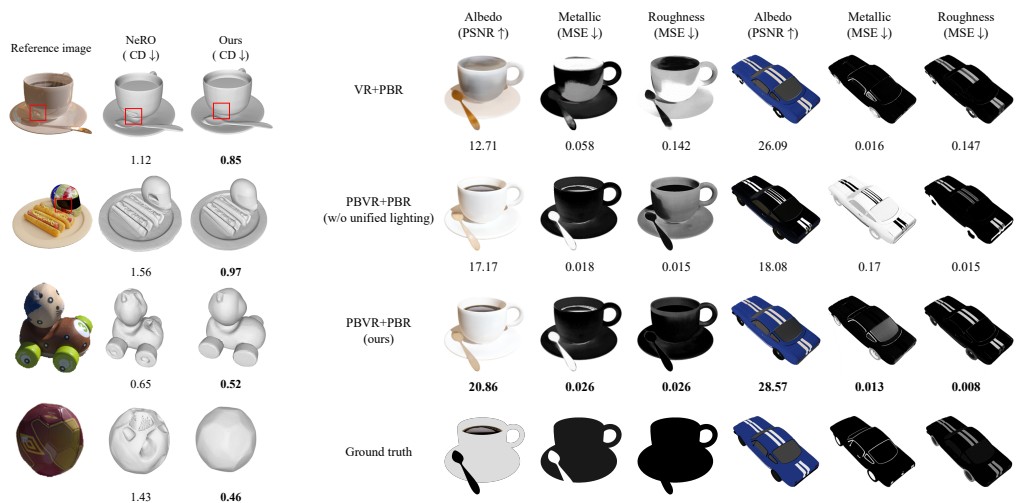

Figure 6: The qualitative comparison of reconstruction between NeRO and our method. Incorrect material estimation hinders the geometry reconstruction.

Figure 7: Ablation study on "coffee" and "car" dataset from ShinyBlender. "VR + PBR" indicates integrating volume rendering and physically based rendering. "PBVR + PBR (w/o unified lighting)" indicates that we used two different light MLPs for PBVR and PBR, respectively. The number below each image indicates the evaluation metric.

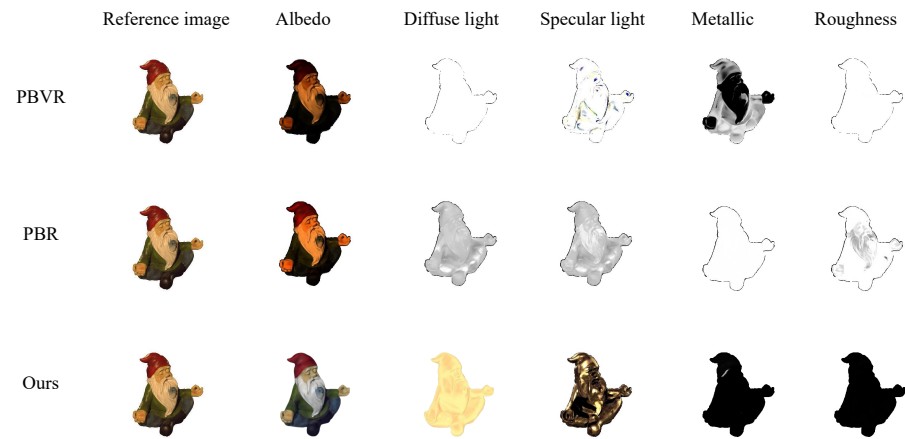

Figure 8: Comparison with the separate use of PBVR and PBR on the "gnome" from the Stanford-ORB demonstrates that our method effectively disentangles lighting and albedo from the appearance.

light representation. We first compared our method with naively integrating traditional volume rendering and PBR. However, traditional volume rendering struggles to reconstruct reflective surfaces accurately, which is essential for optimizing materials and lighting. Consequently, we employed Ref-NeuS Ge et al. (2023) for volume rendering, which demonstrated ideal reconstruction results for specular surfaces. The results are shown in Figure 7 denoted as "VR + PBR". Next, we highlight the importance of unifying lighting representation by encoding the light for PBVR and PBR with two different light MLPs, respectively. The results are shown in Figure 7 indicated as "PBVR + PBR (w/o unified lighting)". Our method significantly improves the performance of materials estimation. We then compared our method with the two-stage optimization approach on the "gnome" object from the Stanford-ORB dataset to validate the effectiveness of unifying PBR and PBVR. In the first stage, PBVR was utilized for surface reconstruction. In the second stage, based on the derived geometry, PBR was applied for materials and lighting estimation. The results, as illustrated in Figure 8, demonstrate that our method effectively disentangles lighting and albedo from appearance, whereas the two-stage approach with PBVR and PBR results in entangled outputs. More visualization of the ablation study can be found in the Appendix.

## 5    LIMITATION AND CONCLUSION

**Limitation.** Although our method shows promising results in inverse rendering across various object types, several limitations remain. Firstly, our approach does not account for shadow effects, which are often incorrectly attributed to the albedo rather than diffuse lighting. Given our method's proficiency in recovering lighting and geometry, we are able to infer shadows based on the geometry and highlights. We plan to explore this capability as part of our future work. Secondly, while PBVR and PBR impose significant constraints on inverse rendering, they can lead to consistent mis-estimations of materials. When they converge on the same incorrect material prediction, the unified framework fails under specific conditions. We show some examples in the Appendix.

**Conclusion.** In this paper, we explore the issue of inverse rendering for various object types, a topic that serves as a critical bridge between NeRF-based models and conventional rendering engines, yet remains under-explored. The inherent ambiguity among geometry, materials, and lighting can significantly hinder accurate decomposition. Our method, Uni-IR, effectively addresses this challenge by integrating physically based volume rendering and physically based rendering into a unified framework. Both rendering methods directly reason materials, lighting and geometry. With a carefully designed unified representations for both lighting and materials, our approach impose mutual constraints and achieve significant performance on inverse rendering task.

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

# A   Appendix

## A   Optimization and Model Details

**Optimization Details.** We employed Adam Kingma & Ba (2014) as our optimizer . Initially, over the first 5,000 iterations, the learning rate was progressively increased from $1 \times 10^{-5}$ to $5 \times 10^{-4}$ via a warm-up strategy. Subsequently, it was reduced to $1 \times 10^{-5}$. For Monte Carlo sampling, we used $N_d = 256$ for the diffuse lobe and $N_s = 512$ for the specular lobe. We sampled 512 rays for each batch. Each ray involved sampling 64 coarse points, 64 fine points, and 32 background points, following the methodology in NeRF++ Zhang et al. (2020). We apply the standard gamma correction to get colors in the sRGB space before computing the rendering loss. For $\mathcal{L}_{\text{vis}}$, given that up to $512 \times N_s$ directions are sampled, computing $v^{\text{march}}$ for all directions incurs substantial computational costs. Consequently, we capped the maximum number of computations for $v^{\text{march}}$ at 8129 for each batch. The weights used in the loss computation are $\lambda_{\text{render}}^{\text{PBR}} = 0.1$, $\lambda_{\text{eik}} = 0.1$, $\lambda_{\text{vis}} = 1.0$ and $\lambda_{\text{mat\_reg}} = 0.5$. Each model was trained over 200,000 iterations, requiring a total of 12 hours on a single NVIDIA RTX 3090 Ti GPU.

**Network architecture.** Our network architecture, akin to NeRO Liu et al. (2023), includes a geometry network, a material network, two light networks, and two visibility networks. These components respectively encode the signed distance function (SDF), material parameters, light representations, and visibility terms. The geometry network, which parametrizes the signed distance function, comprises 8 hidden layers, each with 256 units. Unlike traditional ReLU activation, we employ Softplus with a $\beta$ parameter of 100 across all layers. Additionally, a skip connection Mildenhall et al. (2021) links the input directly to the output of the fourth layer. It takes a spatial position $\mathbf{x}$ as input and outputs signed distances. The material network comprises a material feature network and three dedicated networks for encoding albedo, metalness, and roughness. The material feature network accepts spatial position $\mathbf{x}$, outputting a 256-dimensional material feature vector. This vector is subsequently processed by three networks to predict albedo, metalness, and roughness. The material feature network consists of 8 hidden layers, each with 256 units. Similarly, the networks for albedo, metalness, and roughness each include 4 hidden layers with 256 units. The direct light MLP $l_{\text{direct}}$, indirect light MLP $l_{\text{indirect}}$, and the visibility MLPs $\mathcal{V}^{\text{PBR}}$ and $\mathcal{V}^{\text{VR}}$ each comprise 4 layers, with each layer containing 256 units. For all MLPs that take the position $\boldsymbol{x}$ as input, we utilize positional encoding with a frequency of 8. For MLPs that receive a single direction as input, positional encoding with a frequency of 6 is employed. For both light MLPs, we use exponential function to get light radiance in $[0, \infty)$. For material network, we use Sigmoid to get values in $[0, 1]$.

## B   Evaluation Details

**Geometry.** For the ShinyBlender Verbin et al. (2022) and CompoBlender datasets, the ground truth meshes were exported from the source Blender files. Given that the original models were quite small, with a radius of approximately 1, we applied a scale factor of 150 during export. Similarly, we increased the scale of the reconstructed results by 150 times on the Stanford-ORB dataset, ensuring consistent scales across all ground truth meshes. During training, we normalized the objects to fit within a unit sphere for both the ShinyBlender and CompoBlender datasets. For inference, we transformed the meshes back to their original dimensions to accurately compute the Chamfer Distance. For the Stanford-ORB dataset, we adhered to the protocols defined in the officially released dataset code for training.

Since the original meshes contained too few points, we upsampled the points within each triangle to generate dense point clouds suitable for evaluation. Subsequently, the Chamfer Distance was computed by

$$d(S_1, S_2) = \frac{1}{S_1} \sum_{x \in S_1} \min_{y \in S_2} ||x - y||_2^2 + \frac{1}{S_2} \sum_{y \in S_2} \min_{x \in S_1} ||y - x||_2^2, \qquad (21)$$

where the first term is employed to assess *accuracy*, while the second term evaluates *completeness* Aanæs et al. (2016). Here, $S_1$ and $S_2$ represent the recovered point clouds that are upsampled from the meshes and the ground truth dense point clouds, respectively. For ShinyBlender and CompoBlender, we only reported the *accuracy* as suggested by Ref-NeuS Ge et al. (2023), and for

Standford-ORB, we reported the mean of *accuracy* and *completeness*, since the ground-truth meshes only contrain one-layer surface.

**Materials.** Material estimation involves calculating the Mean Square Error (MSE) for metalness and roughness, as well as the PNSR for albedo. For metalness and roughness, MSE is computed in linear space. Regarding albedo, the predicted values, initially in linear space, are converted to sRGB space for PSNR computation. For the ShinyBlender and CompoBlender datasets, we compared the diffuse albedo, expressed as $(1 - m) * \boldsymbol{a}$, with the ground truth diffuse albedo rendered by Blender. For the Stanford-ORB dataset, we compared the predicted albedo $\boldsymbol{a}$ with the pseudo ground truth albedo. Since there is no ground-truth data for metalness and roughness in real captured datasets, we evaluated the relighting performance instead, using PSNR, SSIM, and LPIPS as evaluation metrics. Improved geometry and materials estimation directly contribute to enhanced relighting results.

**Lighting.** To evaluate the lighting estimation, we utilize three spheres with distinct materials: mirror silver, matte silver, and diffuse grey. The material parameters are defined as follows:

- Diffuse Grey: m = 0.0, $\rho = 0.364$, $\boldsymbol{a} = \{0.8, 0.8, 0.8\}$.
- Matte Silver: m = 1.0, $\rho = 0.327$, $\boldsymbol{a} = \{0.8, 0.8, 0.8\}$.
- Mirror Silver: m = 1.0, $\rho = 0.0$, $\boldsymbol{a} = \{0.8, 0.8, 0.8\}$.

These spheres are rendered using both ground-truth lighting and the estimated environment map in Blender. We assess the performance by calculating the RMSE, si-RMSE, and Angular Error between the two rendered images. To mitigate issues caused by excessively high light intensity in the ground truth environment map, we employ a mask to filter out excessively large light intensities. To acquire the predicted 2D environment map from the direct light MLP, we employed the Equirectangular Projection for mapping a sphere to a rectangle with resolution $512 \times 1024$.

## C  DATASET

**ShinyBlender**  The ShinyBlender dataset, introduced in Ref-NeRF Verbin et al. (2022), aims at the novel view synthesis task for specular surfaces. The original dataset does not include ground truth for diffuse albedo, metallic, and roughness. We re-rendered the dataset using Blender, maintaining consistent camera poses with the original dataset for each object.

**CompoBlender**  We combined individual objects from the ShinyBlender and Blender datasets to create the CompoBlender dataset. This dataset is designed to validate the effectiveness of our method in more complex scenes. First, we combined the "helmet" from ShinyBlender with a part of the "hotdog" from the Blender dataset to create the "hothel" dataset, which features both shiny and diffuse materials. Second, we combined the "toaster" and "helmet" from ShinyBlender to create the "toahel" dataset, which includes indirect lighting and inter-reflections. For rendering multi-view images, we implemented the code from NeRFactor Zhang et al. (2021b). Additionally, we added output nodes for metallic, roughness, and diffuse albedo to evaluate materials. For the environment map, we used the same environment map as "musclecar" for these two scenes.

**Stanford-ORB**  The dataset comprises 14 common objects with different materials captured in 7 natural scenes. For each object, 60 training views and 10 testing views are provided, featuring both high dynamic range (HDR) and low dynamic range (LDR) images under three different scenes. We used LDR images for training and testing. For each object, one scene is selected for training, while the remaining two scenes are used for relighting evaluation. Specifically, we observed that there is always one scene where the object is captured in an outdoor environment. This outdoor scene was consistently chosen for training.

## D  BRDF PARAMETERIZATION

In Sec. 3.1 we introduce the $D$, $F$ and $G$ term of the specular component of BRDF property. We implement the Cook-Torrance BRDF model Cook & Torrance (1982). The basic specular albedo $F_0 = (m * \boldsymbol{a} + (1 - m) * 0.04)$, where $\boldsymbol{a}$ is the albedo and $m$ is the metalness. The Fresnel term (F)

Table 3: Comparison of surface reconstruction with state-of-the-art methods for each object in the ShinyBlender and CompoBlender datasets. **Bold** results have the best score. Chamfer Distance (lower is better) is used as the evaluation metric.

| Methods | ShinyBlender | | | | | | CompoBlender | | |
|---|---|---|---|---|---|---|---|---|---|
| | helmet | toaster | coffee | car | teapot | mean | hothel | toahel | mean |
| NeuS | 0.85 | 3.04 | 1.37 | 0.72 | 0.78 | 1.35 | 1.24 | 1.82 | 1.53 |
| GShader | 0.78 | 2.54 | 1.61 | 0.78 | 1.13 | 1.37 | 1.32 | 2.06 | 1.69 |
| InvRender | 0.68 | 2.34 | 3.24 | 0.58 | 1.03 | 1.39 | 1.02 | 1.67 | 1.35 |
| NvdiffRec | 2.67 | 3.89 | 3.31 | 1.76 | 1.33 | 2.59 | 2.67 | 3.23 | 2.95 |
| NeRO | 0.51 | 0.42 | 1.12 | 0.47 | 0.86 | 0.68 | 1.56 | 2.54 | 2.05 |
| Ours | **0.47** | **0.42** | **0.85** | **0.41** | **0.75** | **0.58** | **0.97** | **0.67** | **0.82** |

is defined as:

$$F = F_0 + (1 - F_0)(1 - (\boldsymbol{h} \cdot \boldsymbol{\omega}_o))^5, \tag{22}$$

where $\boldsymbol{h}$ is the half-way vector between $\omega_o$ and viewing direction $\boldsymbol{\omega}_i$. The normal distribution function $D$ is Trowbridge-Reitz GGX distribution as

$$D(h) = \frac{\alpha^2}{\pi \left((\mathbf{n} \cdot \mathbf{h})^2(\alpha^2 - 1) + 1\right)^2}, \tag{23}$$

where $\alpha = \rho^2$, $\mathbf{n}$ is the surface normal. The geometry term $G$ is the Schlick-GGX Geometry function:

$$G(\boldsymbol{n}, \boldsymbol{\omega}_o, \boldsymbol{\omega}_i, k) = G_{\text{sub}}(\boldsymbol{n}, \boldsymbol{\omega}_o, k)G_{\text{sub}}(\boldsymbol{n}, \boldsymbol{\omega}_i, k), \tag{24}$$

where $G_{\text{sub}}$ is given by:

$$G_{\text{sub}}(\boldsymbol{n}, \boldsymbol{\omega}, k) = \frac{\boldsymbol{n} \cdot \boldsymbol{\omega}}{(\boldsymbol{n} \cdot \boldsymbol{\omega})(1 - k) + k}, \tag{25}$$

where $k$ is a parameter related to the roughness $\rho$, often approximated as $k = \frac{\rho^4}{2}$.

The simplified process from Eq. (8) to Eq. (10) follows the split-sum approximation Karis & Games (2013), where $\boldsymbol{c}_{\text{specular}}$ in Eq. (8) can be rewrited as

$$\boldsymbol{c}_{\text{specular}}(\boldsymbol{x}, \boldsymbol{\omega}_o) \approx \int_{\Omega} L(\boldsymbol{x}, \boldsymbol{\omega}_i)D(\hat{\boldsymbol{d}}, \rho)d\boldsymbol{\omega}_i \int_{\Omega} \frac{DFG}{4(\boldsymbol{\omega}_o \cdot \boldsymbol{n})}d\boldsymbol{\omega}_i, \tag{26}$$

where $\boldsymbol{c}_{\text{specular}}(\boldsymbol{x}, \boldsymbol{\omega}_o)$ is the integral of specular lights on the normal distribution function $D(\hat{\boldsymbol{d}}, \rho)$, where $\hat{\boldsymbol{d}}$ is the reflective directions. The latter part indicates the integral of BRDF, which can be directly computer as $(1 - m) * 0.04 + m * \boldsymbol{a} * F_1 + F_2$, where $F_1$ and $F_2$ are pre-computed scalars and stored in a 2D lookup texture related to $\rho$, $\boldsymbol{n}$ and $\boldsymbol{\omega}_o$. So the Eq. (8) can be modified as

$$\boldsymbol{c}_{\text{specular}}(\boldsymbol{x}, \boldsymbol{\omega}_o) \approx ((1 - m) * 0.04 + m * \boldsymbol{a} * F_1 + F_2) \int_{\Omega} L(\boldsymbol{x}, \boldsymbol{\omega}_i)D(\hat{\boldsymbol{d}}, \rho)d\boldsymbol{\omega}_i. \tag{27}$$

# E  DETAILED RESULTS

We provided detailed quantitative metrics for each individual object in the ShinyBlender, CompoBlender, and Stanford-ORB datasets.

**Geometry reconstruction results.** For geometry reconstruction, we reported the Chamfer Distance (CD) for each object across the three datasets. Table 3 presents the results for ShinyBlender and CompoBlender. Table 4 shows the results comparison with NeRO of geometry reconstruction, relighting and albedo estimation on Stanford-ORB. We further reported the detailed reconstruction comparison with Nvdiffrec, IDR and GaussianShader and NeRO on Stanford-ORB dataset in Table **??**. Note that to make consistent scale with our result on ShinyBlender, we follow the evaluation methodology for CD from Ref-NeuS Ge et al. (2023), so the number is not the same as reported in the website of Stanford-ORB. Our method achieves significant improvement compared to IDR, the best performanced method in Stanford-OBR before. We show the qualitative comparison with GaussianShader on Stanford-ORB in Figure 9 and geometry comparison with NvdiffRec and IDR in Figure 10.

Table 4: Comparison of surface reconstruction with NeRO, GaussianShader, Nvdiffrec and IDR for each object in the Standford-ORB datasets. **Bold** results have the best score.

| Methods | baking | ball | blocks | cactus | car | chips | cup | curry | gnome | grogu | pepsi | pitcher | salt | teapot | mean |
|---|---|---|---|---|---|---|---|---|---|---|---|---|---|---|---|
| | | | | | | Geometry ( CD ↓) | | | | | | | | | |
| Nvdiffrec | 1.76 | 0.89 | 1.96 | 1.35 | 1.62 | 2.59 | 3.29 | 1.89 | 1.72 | 2.02 | 2.71 | 1.92 | 1.46 | 1.32 | 1.89 |
| IDR | 1.18 | 0.50 | 1.49 | 0.82 | 1.07 | 2.01 | 2.77 | 1.41 | 0.86 | **1.60** | 2.04 | 1.53 | 0.82 | 0.85 | 1.35 |
| GShader | 1.07 | 0.49 | 1.05 | 0.87 | 0.65 | 0.99 | 1.99 | 1.39 | 0.81 | 2.01 | 1.71 | 1.92 | 0.66 | 0.72 | 1.22 |
| NeRO | 0.93 | 1.43 | 1.08 | 0.90 | 0.65 | 0.85 | **1.62** | 1.77 | 1.74 | 1.83 | 0.87 | 2.39 | 2.19 | 0.60 | 1.35 |
| Ours | **0.91** | **0.46** | **0.96** | **0.75** | **0.52** | **0.69** | 1.69 | **1.26** | **0.74** | 1.82 | **0.64** | **2.12** | **0.46** | **0.59** | **0.97** |

Table 5: Comparison of materials estimation with state-of-the-art methods for each object in the ShinyBlender and CompoBlender datasets. **Bold** results have the best score.

| | ShinyBlender | | | | | | CompoBlender | | |
|---|---|---|---|---|---|---|---|---|---|
| Methods | helmet | toaster | coffee | car | teapot | mean | hothel | toahel | mean |
| | | | Roughness (MSE ↓) | | | | | | |
| GShader | 0.047 | 0.016 | 0.108 | 0.022 | 0.011 | 0.041 | 0.102 | 0.149 | 0.125 |
| InvRender | 0.045 | 0.014 | 0.098 | 0.011 | 0.008 | 0.035 | 0.089 | 0.048 | 0.069 |
| NvdiffRec | 0.049 | 0.017 | 0.123 | 0.026 | 0.010 | 0.045 | 0.122 | 0.154 | 0.138 |
| NeRO | **0.034** | 0.009 | 0.063 | 0.008 | 0.004 | 0.023 | 0.072 | **0.026** | 0.063 |
| Ours | 0.037 | **0.009** | **0.026** | **0.008** | **0.001** | **0.015** | **0.043** | 0.036 | **0.039** |
| | | | Metalness (MSE ↓) | | | | | | |
| GShader | 0.031 | 0.141 | 0.093 | 0.070 | 0.008 | 0.069 | 0.094 | 0.107 | 0.101 |
| InvRender | - | - | - | - | - | - | - | - | - |
| NvdiffRec | 0.037 | 0.147 | 0.102 | 0.074 | 0.009 | 0.074 | 0.104 | 0.115 | 0.110 |
| NeRO | 0.016 | 0.097 | **0.023** | **0.012** | 0.003 | 0.030 | 0.069 | 0.041 | 0.055 |
| Ours | **0.013** | **0.070** | 0.026 | 0.013 | **0.001** | **0.025** | **0.020** | **0.032** | **0.026** |
| | | | Diffuse Albedo (PSNR ↑) | | | | | | |
| GShader | 15.32 | 18.23 | 17.52 | 23.97 | 25.40 | 20.10 | 16.32 | 15.69 | 16.01 |
| InvRender | - | - | - | - | - | - | - | - | - |
| NvdiffRec | 14.62 | 18.47 | 16.92 | 24.37 | 25.10 | 19.90 | 16.34 | 15.52 | 15.93 |
| NeRO | 16.13 | 21.76 | 19.56 | 27.37 | 26.50 | 22.26 | 17.85 | 17.52 | 17.68 |
| Ours | **16.54** | **22.90** | **20.86** | **28.57** | **26.77** | **23.13** | **18.92** | **18.75** | **18.84** |

**Materials estimation results.** For material estimation, we reported the MSE for metalness and roughness, and PSNR for diffuse albedo for each object in the synthetic datasets, as shown in Table 5. Additionally, we report the relighting performance metrics, including PSNR, SSIM, and LPIPS, as well as PSNR for albedo for each object in the Stanford-ORB dataset, in the middle and bottom parts of Table 4. Note that the albedo is only a pseudo ground truth albedo, which is predicted using NVDiffRec Hasselgren et al. (2022).

**Lighting estimation results.** For lighting estimation, we reported the RMSE, Si-RMSE and Angular Error between two rendered images on three spheres with different materials, which are rendered with ground truth environment map and predicted environment map. The detailed results for each object in ShinyBlender are reported in Table 7. The detailed results for each object in CompoBlender

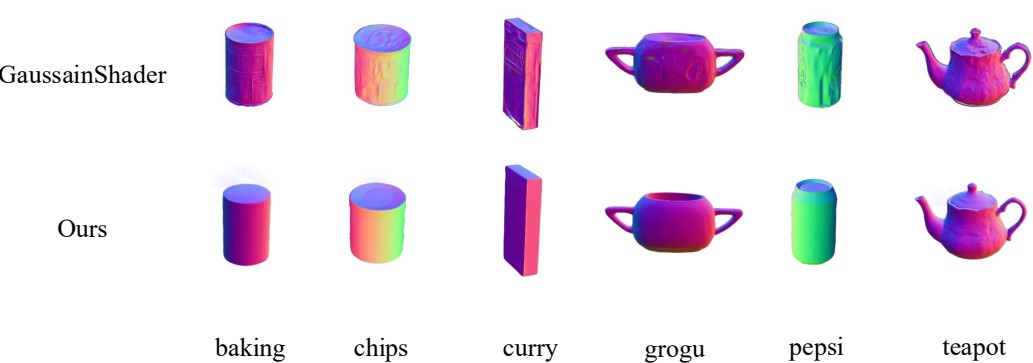

Figure 9: The surface normal of GaussainShader and our method on Stanford-ORB dataset.

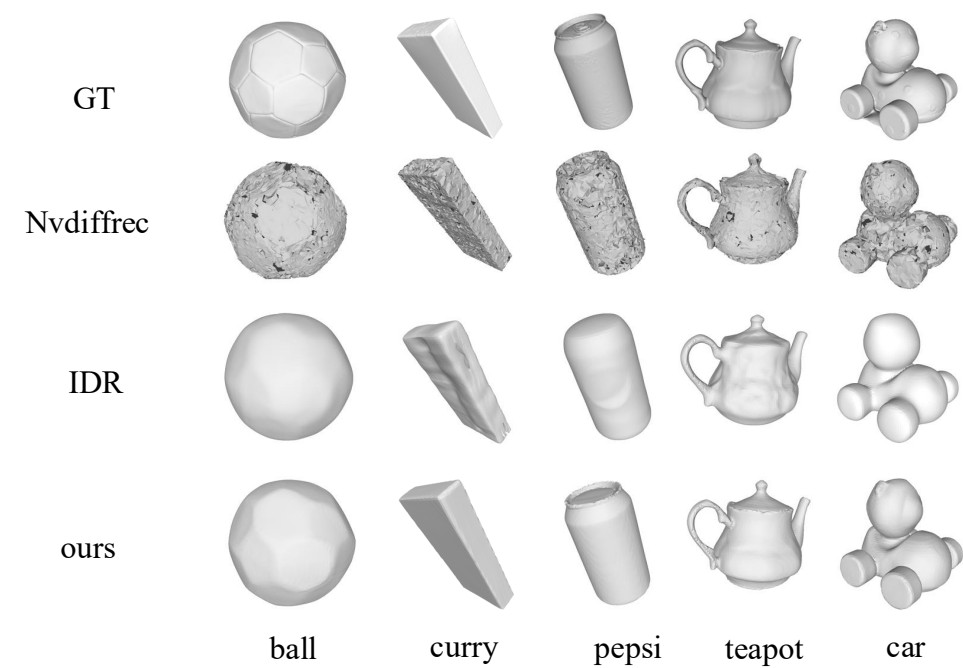

GT

Nvdiffrec

IDR

ours

ball     curry     pepsi     teapot     car

Figure 10: More reconstruction results on objects from Stanford-ORB. IDR, Nvdffrec are compared.

Table 6: Comparison of novel view synthesis with Ref-NeRF and NeRO, including the first stage PBVR and the second stage PBR. **Bold** results have the best score.

| Methods | PSNR ↑ | SSIM ↑ | LPIPS ↓ |
|---|---|---|---|
| Ref-NeRF | 27.86 | 0.878 | 0.375 |
| NeRO (PBVR) | 29.73 | **0.904** | 0.326 |
| NeRO (PBR) | 27.53 | 0.866 | 0.384 |
| Ours (PBVR) | **29.77** | 0.902 | **0.324** |
| Ours (PBR) | 29.51 | 0.894 | 0.329 |

are reported in Table 8. We further reported the detailed relighting metrics and albedo estimation metric on Stanford-ORB dataset compared with NeRO in Table 9.

**Novel-view synthesis quality.** To show the quality of novel view synthesis (NVS), we additionally reported the NVS quality in Table 6 compared to Ref-NeRF, the PBVR and PBR in NeRO and our method in terms of PSNR, SSIM, and LPIPS. The qualitative comparison with NeRO are visualized in Figure 11. The rendered image of "NeRO (PBR)" is inferior to the same PBR rendered image in our unified framework, which shows that our method enhances PBR anti-aliasing capability. The rendered results of "NeRO (PBVR)" and our framework are comparable.

**Relighting quality.** For the relighting evaluation, we selected challenging objects that include inter-reflections. SpeciÌcally, the "cat" from GlossyBlender dataset and the "coffee" from ShinyBlender dataset were chosen. The results of these evaluations are reported in Figure 12.

# F DISCUSSION ABOUT THE RECONSTRUCTION RESULTS.

Here, we discuss why physically based volume rendering fails to reconstruct accurate surfaces, while unifying PBR and PBVR achieves better geometry reconstruction results.

NeuS Wang et al. (2021), an SDF-based neural implicit surface learning method that employs traditional volume rendering, excels at recovering the surfaces of diffuse materials but fails to correctly reconstruct specular surfaces. This failure occurs because the specular color is not well-estimated when the camera view direction conditions the radiance MLP.

Table 7: Comparison of lighting estimation with state-of-the-art methods for each object in the ShinyBlender Dataset. **Bold** results have the best score.

| Methods | helmet | toaster | coffee | car | teapot | mean |
|---|---|---|---|---|---|---|
| | | | Angular Error ↓ | | | |
| GShader | 5.61 / 9.01 / 14.52 | 8.32 / 11.05 / 15.12 | 17.90 / 16.41 / 23.24 | 5.01 / 6.59 / 13.97 | 5.92 / 7.81 / 9.95 | 8.55 / 10.17 / 15.36 |
| InvRender | 5.46 / 8.78 / 13.87 | 8.12 / 10.93 / 14.92 | 18.70 / 16.86 / 23.93 | 4.95 / 6.53 / 13.82 | 5.97 / 7.84 / 10.02 | 8.64 / 10.19 / 15.31 |
| NvdiffRec | 4.92 / 8.15 / 13.45 | 9.15 / 11.58 / 12.98 | 25.65 / 28.46 / 34.84 | 6.80 / 11.93 / 20.04 | 7.10 / 7.24 / 9.39 | 10.72 / 13.47 / 18.14 |
| NeRO | **3.91 / 7.17 / 11.44** | 6.22 / 8.15 / 7.44 | 17.89 / 15.82 / 11.04 | 3.17 / 4.78 / 9.35 | 4.24 / 8.36 / 13.90 | 7.09 / 8.86 / 10.63 |
| Ours | 4.12 / 7.49 / 12.03 | 6.01 / 7.59 / 7.09 | **16.94 / 14.02 / 10.56** | 1.99 / 1.89 / 7.47 | 3.15 / 6.82 / 11.53 | **6.44 / 7.56 / 9.74** |
| | | | Scale-invariant RMSE ↓ | | | |
| GShader | 0.92 / 2.07 / 7.96 | 1.12 / 4.66 / 6.99 | 0.10 / 0.19 / 0.41 | 0.31 / 1.26 / 6.14 | 0.08 / 0.18 / 0.59 | 0.51 / 1.67 / 4.42 |
| InvRender | 0.90 / 1.99 / 7.83 | 0.99 / 4.26 / 6.76 | 0.08 / 0.15 / 0.35 | 0.27 / 1.06 / 5.95 | 0.06 / 0.14 / 0.56 | 0.46 / 1.52 / 4.29 |
| NvdiffRec | 0.89 / 2.46 / 7.34 | 1.28 / 3.98 / 6.94 | 0.11 / 0.19 / 0.39 | 1.08 / 1.76 / 6.28 | 0.20 / 0.64 / 1.02 | 0.71 / 1.81 / 4.39 |
| NeRO | **0.62 / 1.52 / 6.35** | 0.81 / 3.56 / 2.99 | 0.07 / 0.12 / 0.28 | 0.27 / 0.91 / 6.55 | 0.09 / 0.23 / 0.42 | 0.37 / 1.27 / 3.32 |
| Ours | 0.65 / 1.60 / 6.56 | **0.78 / 3.38 / 2.89** | **0.06 / 0.10 / 0.24** | **0.25 / 0.89 / 5.68** | **0.06 / 0.18 / 0.42** | **0.36 / 1.23 / 3.16** |
| | | | RMSE ↓ | | | |
| GShader | 1.31 / 2.51 / 7.71 | 1.82 / 4.61 / 5.12 | 0.58 / 0.51 / 0.69 | 0.59 / 1.19 / 2.73 | 0.17 / 0.62 / 1.09 | 0.89 / 1.89 / 3.47 |
| InvRender | 1.25 / 2.35 / 7.47 | 1.92 / 4.77 / 5.29 | 0.50 / 0.46 / 0.60 | 0.56 / 1.13 / 2.60 | 0.14 / 0.56 / 0.97 | 0.87 / 1.87 / 3.39 |
| NvdiffRec | 1.08 / 2.46 / 7.10 | 1.68 / 4.10 / 3.56 | 0.96 / 0.88 / 1.03 | 0.74 / 1.77 / 3.20 | 0.42 / 0.64 / 1.04 | 0.98 / 1.97 / 3.19 |
| NeRO | 1.05 / 1.87 / 6.76 | 1.90 / 4.07 / 3.31 | 0.30 / 0.27 / 0.55 | 0.52 / 1.01 / 6.60 | 0.10 / 0.52 / 0.71 | 0.77 / 1.55 / 3.59 |
| Ours | **0.91 / 1.84 / 4.45** | 1.85 / 4.01 / 3.21 | **0.25 / 0.21 / 0.51** | **0.36 / 0.93 / 2.10** | **0.06 / 0.44 / 0.64** | **0.69 / 1.49 / 2.18** |

Table 8: Comparison of lighting estimation with state-of-the-art methods for each object in the CompoBlender Dataset. **Bold** results have the best score.

| Methods | hothel | toahel | mean |
|---|---|---|---|
| | Angular Error ↓ | | |
| GShader | 9.91 / 14.29 / 18.55 | 5.41 / 15.74 / 22.67 | 7.66 / 15.02 / 20.61 |
| InvRender | 10.86 / 15.09 / 19.23 | 5.60 / 16.54 / 23.22 | 8.23 / 15.82 / 21.23 |
| NvdiffRec | 6.06 / 10.47 / 15.64 | 6.00 / 10.43 / 15.62 | 6.03 / 10.45 / 15.63 |
| NeRO | 4.20 / 8.45 / 16.77 | 3.37 / 7.00 / 16.58 | 3.79 / 7.73 / 16.68 |
| Ours | **3.89 / 7.15 / 14.92** | **3.01 / 5.98 / 14.08** | **3.45 / 6.57 / 14.50** |
| | Scale-invariant RMSE ↓ | | |
| InvRender | 0.36 / 0.91 / 4.41 | 0.47 / 1.05 / 5.12 | 0.42 / 0.98 / 4.77 |
| InvRender | 0.42 / 1.04 / 4.53 | 0.59 / 1.29 / 6.52 | 0.51 / 1.17 / 5.53 |
| NvdiffRec | 0.54 / 1.16 / 4.65 | 0.57 / 1.16 / 4.64 | 0.55 / 1.16 / 4.65 |
| NeRO | 0.21 / 0.70 / 4.37 | 0.20 / 0.59 / 4.29 | 0.21 / 0.65 / 4.33 |
| Ours | **0.15 / 0.60 / 3.97** | **0.14 / 0.49 / 3.89** | **0.15 / 0.55 / 3.93** |
| | RMSE ↓ | | |
| InvRender | 0.60 / 0.98 / 2.44 | 0.73 / 1.42 / 2.01 | 0.67 / 1.20 / 2.23 |
| InvRender | 0.70 / 1.08 / 3.04 | 0.83 / 1.71 / 4.01 | 0.77 / 1.40 / 3.53 |
| NvdiffRec | 0.62 / 1.28 / 2.35 | 0.62 / 1.25 / 2.32 | 0.62 / 1.26 / 2.34 |
| NeRO | 0.53 / 0.80 / 1.85 | 0.53 / 0.76 / 1.78 | 0.53 / 0.78 / 1.82 |
| Ours | **0.45 / 0.70 / 1.51** | **0.46 / 0.67 / 1.49** | **0.46 / 0.69 / 1.50** |

The radiance and geometry are entangled and mutually affected due to the intrinsic nature of volume rendering, where geometry determines the weights of radiance integration along a ray. If the radiance is not well-estimated, the geometry is also degraded. This conclusion has been demonstrated in Ref-NeRF Verbin et al. (2022) and Ref-NeuS Ge et al. (2023). These methods reparameterize the radiance network as a function of the reflection direction about the surface normal, providing better modeling for radiance and hence improved reconstruction results.

In PBVR, the radiance is influenced by both lighting and materials. Correct lighting and materials also contribute to better geometry, consistent with the aforementioned conclusion. However, with only PBVR, the lighting and materials are often inaccurately estimated, especially in complex or real scenes. In contrast, unifying PBVR and PBR provides better materials and lighting estimation during training. PBR models the lighting using Monte Carlo sampling in a more physically plausible way, while PBVR only approximates the lighting. This comprehensive approach enhances the performance of geometry reconstruction.

## G   MORE VISUALIZATION OF ABLATION STUDY

To validate the importance of our proposed unified light representation, we carried out more ablation studies on the Stanford-ORB dataset and Glossy-Blender dataset to validate the effectiveness of

Table 9: Comparison of surface reconstruction with Nvdiffrec, IDR and GaussainShader for each object in the Standford-ORB datasets. **Bold** results have the best score.

| Methods | baking | ball | blocks | cactus | car | chips | cup | curry | gnome | grogu | pepsi | pitcher | salt | teapot | mean |
|---|---|---|---|---|---|---|---|---|---|---|---|---|---|---|---|
| | | | | | | Relighting ( PNSR ↑) | | | | | | | | | |
| NeRO | 26.03 | 22.48 | 25.65 | 26.72 | 25.36 | **28.58** | 25.11 | **24.44** | 26.05 | 24.44 | 25.31 | **27.06** | **24.06** | 25.05 | 25.45 |
| Ours | **26.31** | **23.58** | **26.24** | **27.61** | **27.23** | 27.88 | **26.23** | 23.54 | **28.54** | **26.35** | **26.06** | 26.75 | 23.78 | **25.76** | **26.13** |
| | | | | | | Relighting ( SSIM ↑) | | | | | | | | | |
| NeRO | 0.909 | 0.854 | 0.894 | 0.910 | **0.912** | 0.919 | **0.905** | **0.894** | 0.847 | 0.904 | 0.907 | **0.905** | **0.880** | 0.925 | 0.898 |
| Ours | **0.902** | **0.867** | **0.918** | **0.920** | **0.921** | 0.904 | **0.921** | 0.886 | **0.878** | **0.914** | **0.905** | 0.901 | 0.868 | **0.929** | **0.902** |
| | | | | | | Relighting ( LPIPS ↓) | | | | | | | | | |
| NeRO | 0.041 | 0.067 | 0.091 | 0.050 | 0.034 | **0.025** | 0.053 | **0.061** | 0.090 | 0.051 | 0.048 | **0.069** | **0.046** | 0.036 | 0.054 |
| Ours | **0.039** | **0.056** | **0.068** | **0.042** | **0.029** | 0.028 | **0.047** | 0.065 | **0.084** | **0.047** | **0.045** | 0.073 | 0.049 | **0.033** | **0.050** |
| | | | | | | Albedo ( PSNR ↑) | | | | | | | | | |
| NeRO | 21.16 | 24.93 | 26.71 | 23.12 | 24.89 | 21.10 | 19.90 | **24.48** | 24.14 | 23.10 | 20.42 | **23.00** | **23.20** | 25.39 | 23.25 |
| Ours | **22.24** | **25.00** | **27.95** | **28.95** | **28.56** | **20.45** | **20.37** | 22.56 | **27.45** | **29.15** | **24.59** | 22.02 | 22.69 | **25.76** | **24.84** |

combining PBVR and PBR for imposing mutual constraints in inverse rendering. The results are shown in Figure 13.

## H   FAILURE CASES

Figure 14 presents the inverse rendering results for the "salt" object from the Stanford-ORB dataset, illustrating a failure case of our method. In this scenario, both PBVR and PBR consistently misesti-mate the material properties, converging on an incorrect metalness value of approximately $m \approx 1.0$. Our method also misestimates the metalness at $m \approx 1.0$. We intend to explore material priors to ad-dress this issue in future work. Leveraging the robust recognition capabilities of Large MultiModal-ity Models (LMMs), we can obtain material priors, as demonstrated in Figure 16. Nevertheless, our approach achieves superior geometry reconstruction quality, and the estimated albedo is more accu-rate, introducing less environmental light into the albedo estimation. Another failure case involves the shadow effect as illustrated in Figure 15, where our method tends to attribute the shadow compo-nent to the albedo rather than diffuse light. Given that our method excels at recovering fine-grained geometry and environment maps, which are critical for determining shadow locations, we plan to further explore this capability in future work.

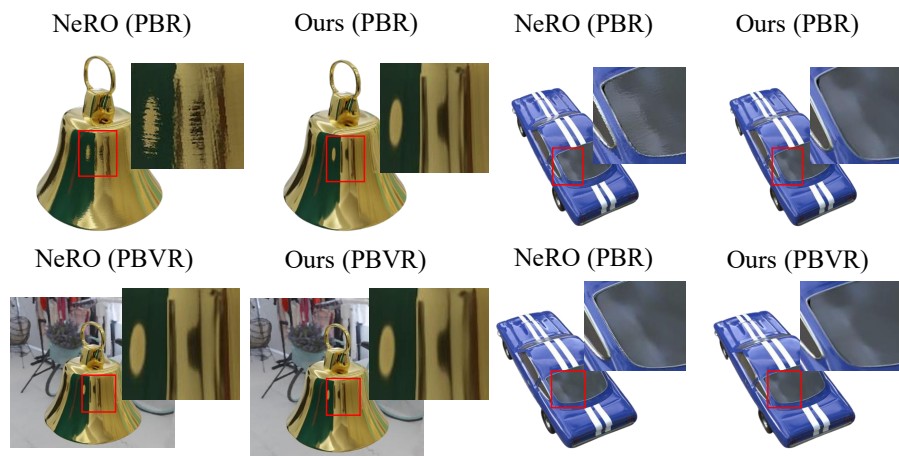

Figure 11: The extracted environment map from the direct light MLP. Our method can recover fine-grained environment map given only multi-view 2D images.

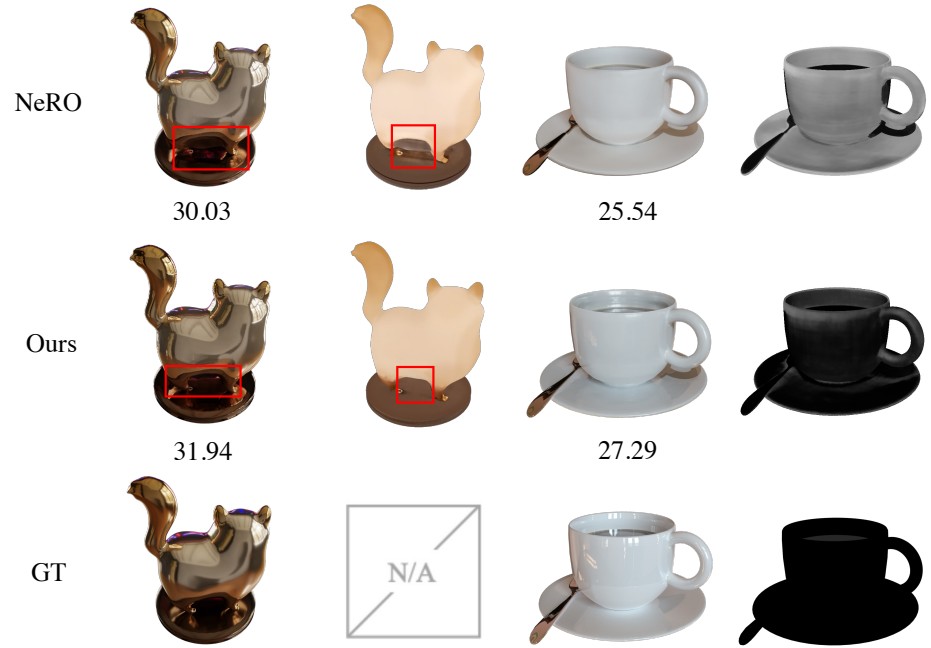

Figure 12: Relighting comparison with NeRO on the ShinyBlender and Glossy-Blender datasets. Our method excels at accurately estimating materials, including albedo (highlighted in the red box in "cat") and roughness ("coffee"), particularly in scenarios with inter-reflection. Correctly recovering these parameters is crucial for accurate relighting.

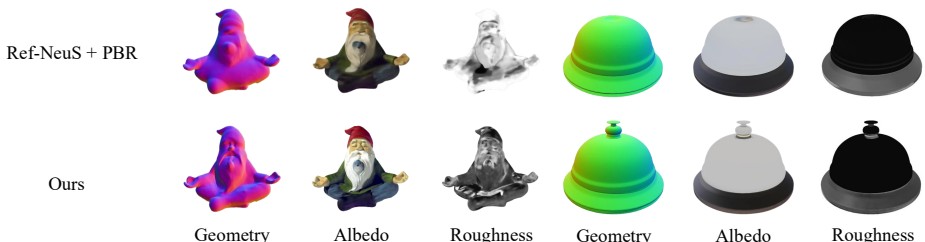

Figure 13: Ablation study on the Stanford-ORB dataset and Glossy-Blender dataset to validate the effectiveness of combining PBVR and PBR for imposing mutual constraints in inverse rendering.

| Reference image | Metallic | Roughness | Normal | Albedo |

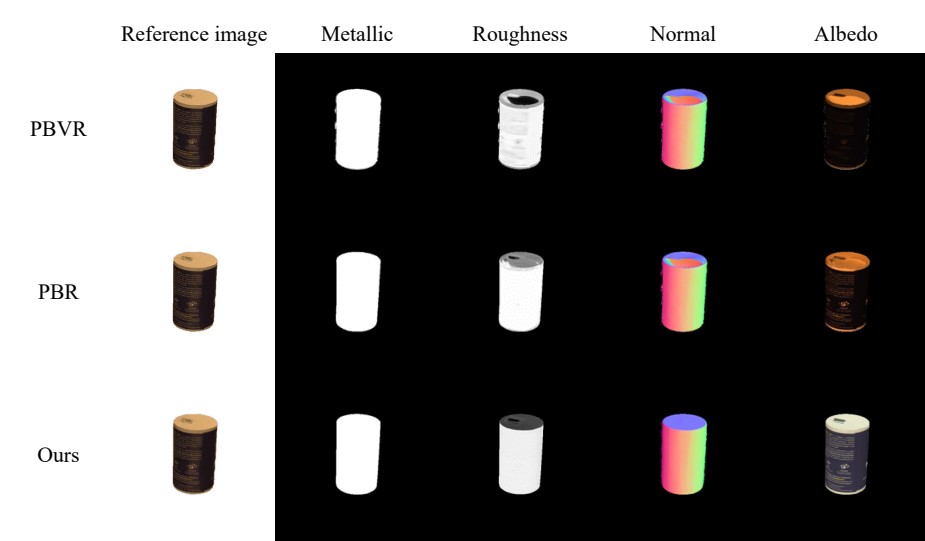

Figure 14: A failure case in our method. When both PBVR and PBR consistently misestimate the material properties, our method also misestimates the material in some circumstances. Nevertheless, our approach achieves superior geometry reconstruction quality.

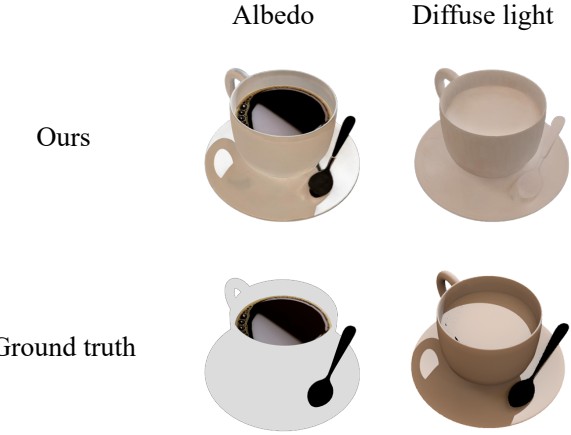

Figure 15: A failure case in our method. The shadow effect are distilled into the albedo instead of diffuse light. The ground truth diffuse light is rendered by Blender.

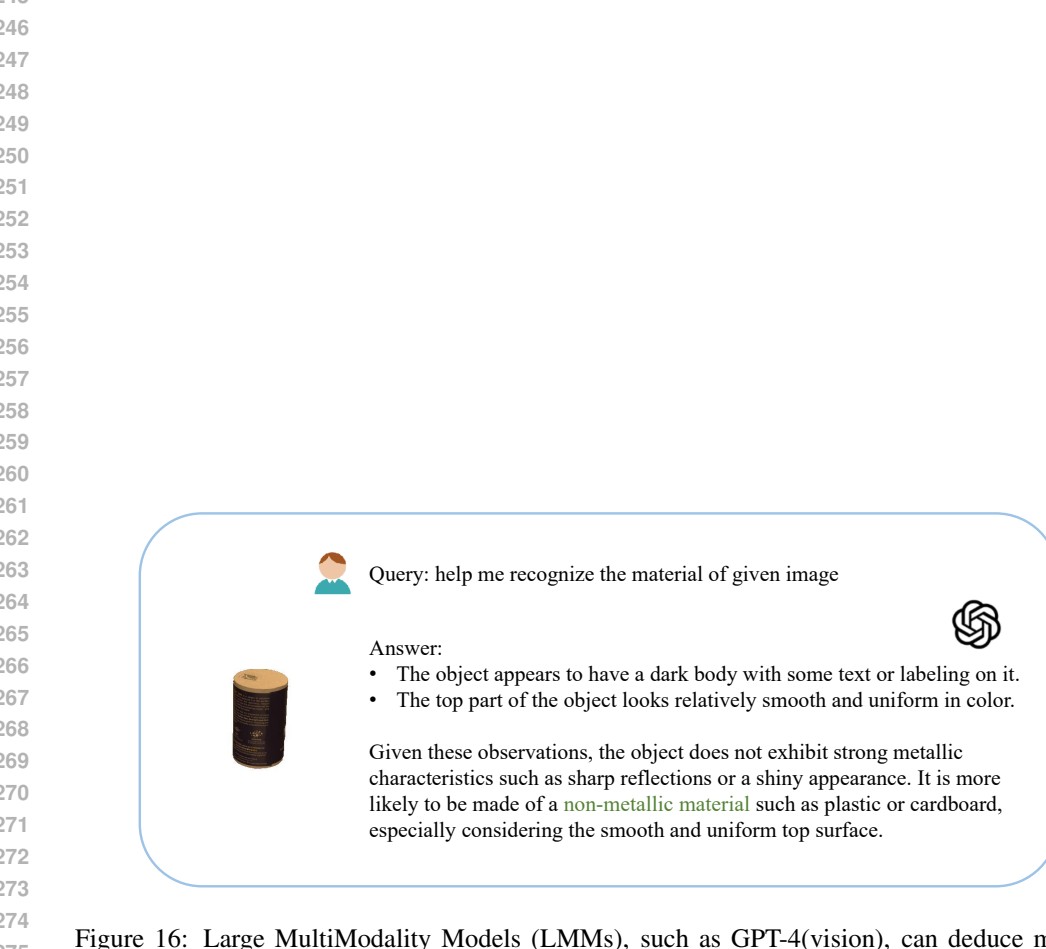

Figure 16: Large MultiModality Models (LMMs), such as GPT-4(vision), can deduce materials from a single image due to their robust recognition capabilities. This material prior can significantly aid in the inverse rendering task.

