# OpenReview forum: "Uni-IR: Ambiguity-Reduced Inverse Rendering through a Unified Framework for Glossy Objects"
_ICLR.cc/2025/Conference — ICLR 2025 Conference Withdrawn Submission_

### Official Review · Reviewer_pZBS · 2024-10-30

**Soundness:** 3
**Presentation:** 2
**Contribution:** 2
**Rating:** 3
**Confidence:** 4

**Summary:**

This paper introduces a method for inverse-rendering the geometry, material, and lighting properties of a scene from captured images. Building on NeRO, it incorporates analytical material reflectance functions and a split-sum light integral to enable material estimation within volume rendering. Additionally, the method proposes a unified approach for joint volume and surface rendering with shared reflectance and lighting networks, achieving superior performance in material estimation and relighting compared to NeRO and other methods.

**Strengths:**

1. The approach of jointly performing volume rendering and surface rendering with shared material and lighting components sounds somewhat new.
2. The paper demonstrates quality improvements over NeRO (which the proposed method is based on) and a few other works.

**Weaknesses:**

1. The paper’s technical contribution is somewhat unclear, as much of the writing focuses on details already introduced by NeRO and other prior works. This makes it a little hard to identify what is new. If understood correctly, 'PBVR' is essentially the same as what NeRO does, and the main new idea in this paper is to combine both PBVR and PBR within the same pipeline. The paper would benefit from better clarification about what is inherited from previous work and what is newly introduced.

2. I am also very concerned about the novelty and technical contribution of this work. Several relevant prior works are not cited, raising questions about the paper’s positioning. Notably, TensoIR and TensoSDF are missing:
    * TensoIR: Tensorial Inverse Rendering, CVPR 2023
    * TensoSDF: Roughness-aware Tensorial Representation for Robust Geometry and Material Reconstruction, SIGGRAPH 2024

     both of which introduce concepts closely related to the proposed method. TensoIR has already combined differentiable volume and surface rendering for inverse rendering, and TensoSDF extends this by integrating NeRO with roughness-dependent IDE, showing superior quality over NeRO. In essence, the proposed approach appears very similar to TensoSDF--which also combines volume and surface rendering based on NeRO--in terms of both high-level ideas and many low-level design choices.

    More specifically, while TensoSDF’s volume rendering focuses on view synthesis (without decomposing material properties), this work aims to achieve material estimation in both volume and surface rendering. However, I am skeptical about whether material decomposition in volume rendering is necessary; without a direct comparison to TensoSDF, it’s hard to determine if this design choice is essential, especially given TensoSDF’s success and advantages over NeRO. Overall, the high similarity to TensoSDF raises serious concerns about the paper’s novelty, technical contribution, and positioning.

3. The paper doesn't include any video results with renderings under changing viewpoints and lighting. Video results are in general critical to evaluate and justify the effectiveness of inverse rendering methods.

**Questions:**

Please check my comments in the weaknesses for my main concerns and questions.

In addition, in Tab 5, what do NeRO (PBVR) and NeRO (PBR) mean? I thought NeRO used PBVR only. I'm more confused about what is new in the work now...

---

### Official Review · Reviewer_b6XX · 2024-11-02

**Soundness:** 3
**Presentation:** 3
**Contribution:** 3
**Rating:** 6
**Confidence:** 3

**Summary:**

The paper proposed a unified framework for geometry, material and lighting reconstruction simultaneously. In contrast to earlier two-stage learning methods, this is an end to end framework making use of mutual constraints for the sake of global optimization. The experiments illustrates that the method outperforms state-of-the-art works.

**Strengths:**

It is reasonable to take advantage of the unified framework for geometry, material and lighting reconstruction, which is potential to derive a better optimal solution. I like the end to end solution, which is able to interact between the two components to optimize globally.

**Weaknesses:**

The paper is not well written. The writing needs more proof read. There are typos, grammar issues and even "?" in the Appendix. For example:

many "Eq. equation"

Line 261: where $i$ indicate the $i$-th sampled direction --> where $i$ indicates the $i$-th sampled direction

Line 913: ??

The real images, e.g., Figure 8, are not natural. It looks still like a synthetic image. Why to choose such images? Can you evaluate on more challenge real images? I suggest to use more natural real images with background for evaluation.

**Questions:**

What metric is used in Table 1? This is not explained clearly. I suggest to clarify these in the Table caption or the text.

I am curious why the image backgrounds are all white. What effect does the background have on the reconstruction?

Since we don't know the surface in prior, how to "evaluate the rendering equation on the surface points $x_s$" in Line 255. Can you give more details on how to identify the surface?

---

### Official Review · Reviewer_BAGv · 2024-11-03

**Soundness:** 3
**Presentation:** 3
**Contribution:** 3
**Rating:** 5
**Confidence:** 4

**Summary:**

The paper presents a novel method, Uni-IR, designed for inverse rendering tasks.

Many existing inverse rendering works employ a two-stage optimization for scene decomposition, estimating geometry and materials/lighting separately. This two-stage manner can lead to suboptimal results due to the ambiguity arising from the interactions between geometry and materials/lighting. Instead, Uni-IR employs a joint inverse rendering technique that integrates physically-based rendering (PBR) with volumetric rendering in radiance fields, effectively addressing the inherent ambiguity between geometry and materials/lighting.

The key contributions of this work include:

1. The first unified optimization framework that seamlessly integrates PBVR (physically-based volume rendering) and PBR.
2. Qualitative and quantitative experiments demonstrating the effectiveness of the method in decomposing geometry, materials, and lighting.

**Strengths:**

### 1. Seamless Integration

The paper tackles the problem of seamlessly integrating physically-based volume rendering (PBVR) with physically-based rendering (PBR). Existing inverse rendering methods rarely unify the two; they either rely on deterministic PBR (Nvdiffrec-like) or use per-sample/per-Gaussian PBR color composition, referred to as "physically-based volume rendering" in NeRF-based methods (e.g., TensoIR, NeRO) and 3DGS-based methods (e.g., GS-IR, Relightable 3DGS, Gaussian Shader). However, directly applying PBR on radiance fields, where the geometry is stochastic, often results in artifacts (see Fig. 11 in the paper, NeRO with PBR).

Although the paper does not delve deeply into the underlying issues, it offers valuable insights by addressing the problem through joint optimization of PBR and PBVR losses.

### 2. Performance

The paper provides extensive experiment results, demonstrating Uni-IR outperforms its baselines a lot in the inverse rendering task.

### 3. Quality & Clarity

The quality of the paper is commendable, providing rich details that include both quantitative and qualitative results, thorough formulations, implementation descriptions, etc.
The clarity of the paper is fair.

**Weaknesses:**

1. The qualitative results primarily focus on lighting reconstruction (Fig. 4; Fig. 5), while material decomposition is only included in the ablation study (Fig. 6), lacking comparison with baseline methods. Also, the results may lack more geometry comparisons with baselines other than NeRO (Fig. 7). Although this is not a major issue, it may diminish the overall soundness of the claims.

2. In Fig. 7, Uni-IR bakes shadow into the albedo. Whether it is a failure case or a common case, may require further clarification (e.g. by visualizing visibility term or showing more albedo results).

**Questions:**

1. The term "unified" in the claims lacks clarity. It could refer to:
(a) a unified optimization v.s. a two-stage optimization, or
(b) unified rendering (PBVR + PBR) v.s. separate renderings (PBVR; PBR).

2. Uni-IR seems to be able to extract mesh as it follows NeuS to build SDF fields. Can this approach finally produces mesh with textured? How about the quality?

3. Does the term "seamless integration" means that, during training, the PBR color and the PBVR color will converge to the same value?

---

### Official Review · Reviewer_SBVP · 2024-11-04

**Soundness:** 3
**Presentation:** 3
**Contribution:** 3
**Rating:** 6
**Confidence:** 4

**Summary:**

The paper presents Uni-IR, a unified framework for inverse rendering that integrates Physically Based Volume Rendering (PBVR) and Physically Based Rendering (PBR) to simultaneously estimate geometry, materials, and lighting in 3D scenes. Unlike traditional two-stage methods that separately optimize these components, Uni-IR employs a joint optimization approach, enhancing consistency and reducing ambiguities between materials and lighting. The framework is evaluated on both synthetic and real-world datasets, demonstrating superior performance in reconstructing accurate geometry, realistic material properties, and precise lighting conditions. Key contributions include the development of a shared representation for materials and lighting across PBVR and PBR, and a novel methodology for comprehensive scene reconstruction that improves fidelity and resolves common inverse rendering challenges.

**Strengths:**

The paper’s strengths lie in its unified approach to estimating geometry, materials, and lighting simultaneously, which enhances consistency and reduces ambiguities between these components. It integrates Physically Based Volume Rendering (PBVR) with Physically Based Rendering (PBR) for a comprehensive optimization process, leading to more accurate and realistic scene reconstructions. The framework’s shared material and lighting representation contribute to robust and coherent outputs. Additionally, the evaluation on both synthetic and real-world datasets highlights the framework’s versatility and superior performance compared to traditional methods.

**Weaknesses:**

The framework’s approach, while powerful, could also be computationally intensive, requiring substantial hardware resources for effective training and rendering.

A major concern of the paper is reproducibility, as the complex method involves many integrated components, yet no code or reproducibility statement is provided, making it difficult for others to replicate the results.

It seems that the lighting estimation experiment (line 364-372) uses known materials rather than having all components estimated simultaneously.

**Questions:**

In each evaluation, do you estimate all scene parameters (geometry, materials, and lighting) simultaneously, or do you assume the other parameters are given when separately estimating each one of these parameters? This could significantly change my rating.

Line 913 is missing table reference.

---

### Note · Authors · 2024-11-13

I have read and agree with the venue's withdrawal policy on behalf of myself and my co-authors.